# Co-distilled attention guided masked image modeling with noisy teacher for self-supervised learning on medical images

**Jue Jiang**[1]                                                                JIANGJ1@MSKCC.ORG
**Aneesh Rangnekar**[1]                                                RANGNEA@MSKCC.ORG
**Harini Veeraraghavan**[1]                                        VEERARAH@MSKCC.ORG
[1] *Memorial Sloan Kettering Cancer Center*

**Editors:** Accepted for publication at MIDL 2025

## Abstract

Masked image modeling (MIM) is a highly effective self-supervised learning (SSL) approach to extract useful feature representations from unannotated data. Predominantly used random masking methods make SSL less effective for medical images due to the contextual similarity of neighboring patches, leading to information leakage and SSL simplification. Hierarchical shifted window (Swin) transformer, a highly effective approach for medical images cannot use advanced masking methods as it lacks a global [CLS] token. Hence, we introduced an attention guided masking mechanism for Swin within a co-distillation learning framework to selectively mask semantically co-occurring and discriminative patches, to reduce information leakage and increase the difficulty of SSL pretraining. However, attention guided masking inevitably reduces the diversity of attention heads, which negatively impacts downstream task performance. To address this, we for the first time, integrate a noisy teacher into the co-distillation framework (termed DAGMaN) that performs attentive masking while preserving high attention head diversity. We demonstrate the capability of DAGMaN on multiple tasks including full- and few-shot lung nodule classification, immunotherapy outcome prediction, tumor segmentation, and unsupervised organs clustering.

**Keywords:** Attention guided masked image modeling, Swin, noise regularized co-distillation.

## 1. Introduction

Self-supervised learning (SSL) is an approach to extract useful feature representations from large cohorts of unlabeled images in a task agnostic way, following which the model can be applied to downstream tasks with minimal fine-tuning. Masked image modeling (MIM) is a highly effective SSL task that extracts useful feature representations from degraded images (He et al., 2021; Zhou et al., 2022; Jiang et al., 2022). MIM fosters token diversity across attention heads and increases locality inductive bias for transformers, which enhances feature reusability for downstream tasks (Xie et al., 2023). MIM is often implemented using random (Zhou et al., 2022; Jiang et al., 2022) or blockwise (Bao et al., 2022) masking of image patches. However, such methods are unsuitable for medical images due to the strong semantic correlation of spatially adjacent patches, which risks information leakage and simplification of the SSL task. Recent work has focused on making MIM challenging by selectively masking discriminative regions. Strategies include masking high-attending (Kakogeorgiou et al., 2022; Liu et al., 2023b), low-attending (Li et al., 2021), as well as semantically relevant (Li et al., 2022; Shi et al., 2022) and hard to reconstruct regions (Wang et al., 2023) derived from auxiliary networks for natural images. The impact of attention guided

MIM has not been studied for medical image analysis. Hierarchical shifted window (Swin) transformers enable multi-scale attention with linear complexity, making them ideal for analyzing volumetric medical images (Tang et al., 2022). Hybrid Swin-convolutional segmentation models combining Swin encoder with convolutional decoders have demonstrated higher accuracy compared to ViT and convolutional networks (Tang et al., 2022; Jiang et al., 2024; Cao et al., 2022). However, attentive masking cannot be performed with Swin due to two key architectural limitations. First, Swin uses windowed attention that limits attention to local regions. Second, Swin lacks the classifier token [CLS] that interacts with all the patch embeddings via self-attention to extract global attention required for attention guided masking. Hence, we introduced an architectural enhancement to Swin for performing attention guided MIM. Our enhancement is a semantic attention (SA) module, consisting of vision transformer (ViT) blocks implemented into Stage#3 with a [CLS] token interact with token embeddings across the whole image. Similar to AttMask (Kakogeorgiou et al., 2022), an exponentially moving average (EMA) teacher guides the masking applied to a co-distilled student network. Unlike AttMask, our approach works for Swin and ViT. Furthermore, to enhance attention diversity (Arani et al., 2021), we introduced a noisy teacher into the co-distillation learning framework by applying patch dropout to the teacher's input. We call our approach that performs co-Distillation Attention Guided MAsking with Noisy teacher, DAGMaN. We show DAGMaN produces global self-attention and achieves higher attention head diversity. Our contributions are: (i) architectural enhancement of Swin to perform attentive masking, (ii) noisy teacher regularized co-distillation for enhancing attention-head diversity, (iii) analysis of the impact of our DAGMaN approach on multiple medical image analysis tasks including lung nodule classification, segmentation of nodules and malignant tumors, lung cancer treatment response prediction, and unsupervised clustering to distinguish organs. All the code and model checkpoints will be made available upon manuscript acceptance.

## 2. Related works

**Masking strategies for MIM :** Random masking (He et al., 2021; Jiang et al., 2022; Xie et al., 2022; Zhou et al., 2022) is a commonly used MIM approach due to its computational simplicity. A key limitation of this approach is the potential of exposing highly correlated and spatially adjacent patches, which simplifies SSL and reduces downstream task accuracy. Block-wise matching (Wei et al., 2022; Bao et al., 2022; Wang et al., 2022) mitigates this issue by masking spatially adjacent patches. Masked regions are still selected randomly, dispersing attention towards irrelevant regions, and wasting computations for pretraining. Attentive masking methods overcome the aforementioned issues by identifying and masking discriminative and spatially coherent regions. Semantic attention (Li et al., 2022) and hard patch selection methods (Wang et al., 2023) use auxiliary networks to identify discriminative regions that add computational complexity and need for different pretrained networks. ViT-based methods use the spatial attention maps computed using the [CLS] token to identify regions for masking, providing a computationally simple approach (Li et al., 2021; Kakogeorgiou et al., 2022; Liu et al., 2023b). However, attention guided masking with [CLS] is not possible to perform with Swin, which our approach overcomes.

**Noisy teacher co-distillation :** Noise injection is an implicit data augmentation method (You et al., 2022) that improves data utilization with reduced memory requirements and improves

Figure 1: DAGMaN: (A) shows the student network with input produced using attention guided masking. (B) shows the noisy teacher that is provided noisy tokens produced using patch dropout. (C) shows the attention guided masking produced using global attention mask *Att* produced by the teacher. (D) depicts the semantic attention block with [CLS] token implemented into Swin in order to extract *Att*.

training convergence (Liu et al., 2023a). Feature dropout-based noise injection has been used for knowledge distillation applied to few or feature layers using same (Lee et al., 2023) as well as different teacher and student models for classification tasks (Arani et al., 2021; Liu et al., 2021a; Bulò et al., 2016). Distillation with noise injected into intermediate outputs or features has shown to enhance accuracy for image-level (Tarvainen and Valpola, 2017; Laine and Aila, 2017) and dense-pixel prediction tasks (You et al., 2022). Different from prior methods, we inject noise into teacher's inputs to increase the variability of extracted features by the two networks, that in turn enhances attention head diversity.

## 3. Method

Swin cannot extract global attention because it conducts self-attention locally within defined spatial windows and shifted window partitioning to maintain cross-window relationships. We introduced semantic attention (SA) block into Swin to extract global attention for attention-guided masking and a noisy teacher co-distillation for attention diversification.

### 3.1. Semantic attention guided masked image modeling

SA (Figure 1 D) is composed of 2 ViT blocks, with each block consisting of a multi-head attention layer followed by a Multi-Layer Perceptron Layer (MLP). A [CLS] token $z_{[CLS]} = z_{N+1}$ is introduced at SA, and interacts with all $i \in N$ features $\{z_i\}_{i=1}^N$ and creates a tokenized feature $z \in \mathbb{R}^{(N+1) \times D}$, where $D$ is the feature size. The multi-head mechanism using $h$ attention heads and individual projection matrices $W_q$, $W_k$, $W_v$ extracts various representations using query $Q^{[CLS]} = W_q \times z_{[CLS]} + b_q$, key $K = W_k \times z + b_k$, and value

$V = W_v \times z + b_v$ features, where $b_q, b_k, b_v$ are the biases. The semantic attention (S$_{ATT}$) averages the representations as:

$$S_{ATT} = \frac{1}{h} \sum_1^h \text{Softmax}(Q_h^{[CLS]} \cdot \frac{K_h^T}{\sqrt{D/h}}).  \tag{1}$$

$S_{ATT} \in [0,1]$ is an attention vector of the same size $N$ as the number of patches, with larger values indicating higher attention. Thus, S$_{ATT}$ is sorted in descending order to identify the top $[rN]$ patches, $r \in [0, 1]$ being the masking ratio, default to 0.7, and setting those indices in mask vector $m^{satt}$ to 0. Inputs to $f_s(\theta_s)$ is produced by $\tilde{u} = m^{satt} \odot u$ (Figure 1 C). Masking all the high-attending tokens can remove the most discriminatory regions, potentially making the SSL task very challenging and hamper convergence. Hence, a small number $s < r$ (default of 0.1), of the top high attending or hint tokens are left unmasked. Swin adopts a hierarchical design, as a result of which features extracted at earlier stages (#1 and #2) may not contain sufficient global context. On the other hand, features extracted after stage #4 have very low resolution. Hence, stage #3 was selected for SA module to balance network depth and attention density.

### 3.2. Attention guided Co-distillation framework with Noisy teacher

Two different cropped views ($u$ and $v$) from the input 3D CT volume are converted into a sequence of $N$ patch tokens $\{u_i\}_{i=1}^N, \{v_i\}_{i=1}^N$ and then subjected to guided and random masking to create corrupted input patch tokens $\tilde{u}$ and $\hat{v}$ for student ($f_s(\theta_s)$ and teacher ($f_t(\theta_t)$) networks, respectively (Figure 1). The networks' parameters are updated using co-distillation losses consisting of attention guided image token distillation (AITD), attention guided masked patch token distillation (AMPD), global image token distillation (GITD). The attention guided masked image prediction (AMIP) is used only for the student model. $L_{AITD}$ measures dissimilarity of the token embeddings with [CLS] and optimizes SA. $L_{AMPD}$ measures the dissimilarity of patch token embeddings extracted by teacher and student networks, emphasizing dissimilarities in the attention guided masked regions. AITD and AMPD losses are computed after stage #3 to take advantage of the denser feature representation compared to stage #4 and higher anatomic context than earlier stages. $L_{GITD}$ that measures the dissimilarity of the token embeddings computed for the whole image views and $L_{AMIP}$ that measures difference in the generated image patches in the masked regions are computed at the end of stage #4.

**Noisy teacher regularization** The teacher network (Figure. 1B) uses an identical architecture as the student and is created using exponential moving average as $\theta_t = \lambda_m \theta_t + (1 - \lambda_m)\theta_s$, where $\lambda_m$ is momentum. Diversity of attention head is increased by using patch dropout (Liu et al., 2023a) to create noisy input tokens $\hat{u}, \hat{v}$ from $u, v$ for the teacher, thus termed noisy teacher. Patch dropout randomly sets elements within mask tokens $m_u, m_v$ corresponding to indices of patch tokens $u, v$ to 0 using a patch drop ratio $r_t \in [0, 1]$, where $r_t = 0.7$. The corrupted inputs $\hat{u} = m_u \odot u$ and $\hat{v} = m_v \odot v$ are thus generated for $f_t(\theta_t)$.

**Attention guided image token distillation** loss measures the dissimilarity of the token distributions $P_s^{[CLS]}(\tilde{u}, \theta_s)$ and $P_t^{[CLS]}(\hat{v}, \theta_t)$, computed for $\tilde{u}, \hat{v}$, produced by linear projection layers placed after the SA blocks in the student $f_s(\theta_s)$ and teacher $f_t(\theta_t)$ networks:

$$L_{AITD} = -\sum_{i=1}^N P_t^{[CLS]}(\hat{v}_i, \theta_t) log(P_s^{[CLS]}(\tilde{u}_i, \theta_s)).  \tag{2}$$

The sharpness of the token distribution is controlled by computing a sharpening transformation (Jiang et al., 2022) using separate temperature terms $\tau_s > 0$ and $\tau_t > 0$ for the student and teacher networks. **Attention guided masked patch distillation** loss measures the dissimilarity of masked token distributions $P_t^{Patch}(\hat{u}, \theta_t)$ and $P_s^{Patch}(\tilde{u}, \theta_s)$, produced from the different corrupted versions ($\hat{u}$ from patch dropout, $\tilde{u}$ from attention guided masking) of same view $u$ to increase robustness of network to image noise. The token distributions are computed by applying *softmax* operation to the stage #3 output of $f_t(\theta_t)$ and $f_s(\theta_s)$, followed by sharpening transformations and averaged on the $m_i^{att}$ tokens. AMPD is computed as:

$$L_{AMPD} = -\sum_{i=1}^{N} m_i^{att} \cdot P_t^{Patch}(\hat{u}_i, \theta_t) log(P_s^{Patch}(\tilde{u}_i, \theta_s)). \tag{3}$$

**Global image token distillation** loss measures the dissimilarity of the global image token distributions $P_s^{[g]}(\tilde{u}, \theta_s)$ and $P_t^{[g]}(\hat{v}, \theta_t)$ produced from the average pooling layer placed after Stage #4, computing the mean of all embeddings in the spatial dimension, producing a global feature representation of the student and teacher networks, respectively. $I_{GITD}$ is then computed as:

$$L_{GITD} = -\sum_{i=1}^{N} P_t^{[g]}(\hat{v}_i, \theta_t) log(P_s^{[g]}(\tilde{u}_i, \theta_s)). \tag{4}$$

The token distributions are subjected to sharpening transforms as used for AITD.

**Attention guided masked image prediction** measures the image reconstruction error between the reconstructed image produced by $f_s(\theta_s)$ using a 1-layer linear projection layer $h_s^{Pred}$ applied to masked input $\tilde{u}$ as:

$$L_{AMIP} = \sum_{i}^{N} E \| m_i^{att} \cdot (h_s^{Pred}(f_s(\tilde{u}_i, \theta_s)) - u_i) \|_1, \tag{5}$$

where $m_i^{att}$ is the masked token vector produced through attention guided masking. The total loss is computed as, $L_{total} = L_{AMIP} + \lambda_{AMPD} L_{AMPD} + \lambda_{AITD} L_{AITD} + \lambda_{GITD} G_{ITD}$.

## 4. Experiments and Results

### 4.1. Datasets

**Pretraining:** SSL pretraining was performed on 10,412 unlabeled 3D CTs (1.89M images) sourced from public (Xiao et al., 2023; Heller et al., 2019; Ji et al., 2022) and institutional cohorts involving diseases in the head and neck, chest, and abdomen (Jiang et al., 2024).
**Downstream tasks and data:** A total of 4,746 cases sourced from three public (Armato III et al., 2011; Aerts et al., 2015; Yang et al., 2023) and one institutional dataset (Jiang et al., 2024) were evaluated for lung nodule classification (Task 1) using lung image database consortium (LIDC) (Armato III et al., 2011), lung nodule and tumor segmentation (Task 2), immunotherapy response prediction (Task 3) (Zhu et al., 2023), and unsupervised clustering of multiple organs using pretrained features (Task 4). Description of the various datasets and fine-tuning details are included in Supplementary Section B.

## 4.2. Experiments

Implementation details for the various networks and analyzed tasks are provided in the Supplemental Section A. Experiments evaluated the benefit of attention guided masking and noisy teacher regularization with Swin. Hence, DAGMaN was compared against SMIT (Jiang et al., 2022), which uses random masking with an identical Swin backbone as well as blockwise masking as used in iBot (Zhou et al., 2022). ViT-based AttMask (Kakogeorgiou et al., 2022) as well as low attending masking used in masked self-supervised transformer (MST) (Li et al., 2021) were also analyzed. All networks were pretrained using identical pretraining, fine-tuning, and testing datasets. Few-shot performance using 25% and 50% cases was used to assess classification (Task 1, Task 3) and segmentation (Task 2). Representative CNN baselines for segmentation, nnU-Net(Isensee et al., 2021) and classification, 3D-ResNet 50(Chen et al., 2019) were used for lung nodule segmentation as well as lung nodule malignancy and immunotherapy outcome prediction, respectively.

## 4.3. Image-level classification performance (Task 1 and 3)

Image-level classification performance was evaluated by using linear probing (LP) with one projection head and fine-tuning (FT) by training all the layers for both Task 1 and 3. DAGMaN was more accurate than other methods, resulting in higher AUC using both FT and LP (Figure 2). Few-shot training showed higher accuracy with DAGMaN than other methods even with 25% training data. 3D ResNet was the second best method for predicting lung nodule malignancy even at but produced worst accuracy with 25% training data for predicting immunotherapy outcomes. DaGMaN had higher specificity and sensitivity compared to other methods for nodule malignancy classification (Task 1) but similar specificity as other methods for predicting immunotherapy outcome (Task 3). The attention maps computed for two different patients with LP and FT using DAGMaN clearly show localization within the tumor (Figure 2 (e)).

## 4.4. Dense-pixel prediction as tumor segmentation (Task 2)

The segmentation network was implemented by adding a U-Net decoder, initialized from scratch, to the pretrained encoders and all networks subjected to FT (Aerts et al., 2015) (training with 350 and validation with 27 cases). Results of testing the model with LIDC and immunotherapy datasets are shown for full-shot (Figure 3 (a,b)) and few-shot (Figure 3 (c)) regimes. DAGMaN outperformed all methods in both full- and few-shot regimes and showed clear performance improvement for the nodule segmentations compared to segmenting larger lung tumors. nnUnet was less accurate than all except MST under all analyzed training data settings, indicating the importance of pretext task in SSL pretraining. It was also similar in performance to AttMask, which uses a ViT architecture. Segmentations for a representative case is shown in Figure 3 (d).

## 4.5. Impact of attentive masking and noisy teacher co-distillation

Diversity of attention heads measured using entropy of attention distances for all the 4 Swin stages is shown in Figure 4 (a). Random masking as performed in SMIT resulted in a high entropy with a small variance in the first two stages but lower entropy with increasing

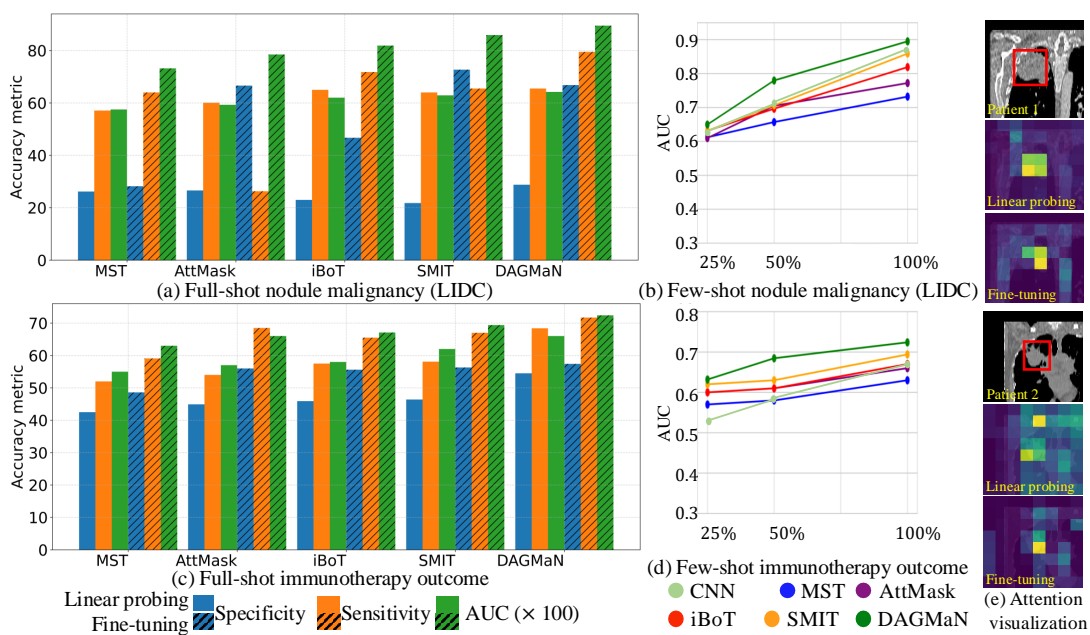

Figure 2: Accuracy comparison of various methods including CNN-based 3D ResNet-50 on two different classification tasks using full-shot (a,c) and few-shot (b,d) training regimes. (e) shows attention maps computed with LP and FT for two representative lung cancer patients, who did not respond to treatment.

variance in the last two stages. Using only semantic attention (SA) reduced average entropy and variance in stages 1 and 4, indicating reduced diversity. On the other hand, adding noisy teacher (NT) increased entropy and variance even for random masking (stages 2 and 3). Similarly, DAGMaN that uses both SA and NT produced high entropy in stages 1, 2, and 3 with higher variance than other methods indicating higher attention diversification. The impact of attention diversification was analyzed for lung nodule malignancy classification (Figure 4 (b)), which showed an overall higher AUC with DAGMaN compared to all other methods. Training using only SA or NT produced substantially lower specificity than DAGMaN, indicating that combining NT and SA improves accuracy.

UMAP clustering (McInnes et al., 2018) performed using features computed from the pre-trained models on the OrganMNIST3D dataset showed the best separation of various organs (Task 4) with DAGMaN (Figure 4 (c)) with highest inter-class and lowest intra-class distance (Supplemental Figure 7). SA guided masking improved organ separation compared to random masking (SMIT) and NT, but was less effective than DAGMaN.

### 4.6. Ablation tests

Analysis of various pretraining losses for nodule malignancy classification (Task 1) showed that removing attentive masked image prediction (AIMP) resulted in the largest accuracy drop (AUC of 0.829) followed by attentive masked patch distillation (AMPD) with AUC of 0.865 (Supplemental Table 1). GITD least impacted accuracy indicated by a small drop in AUC of 0.878. However, inclusion of GITD with other losses produced best AUC of 0.895.

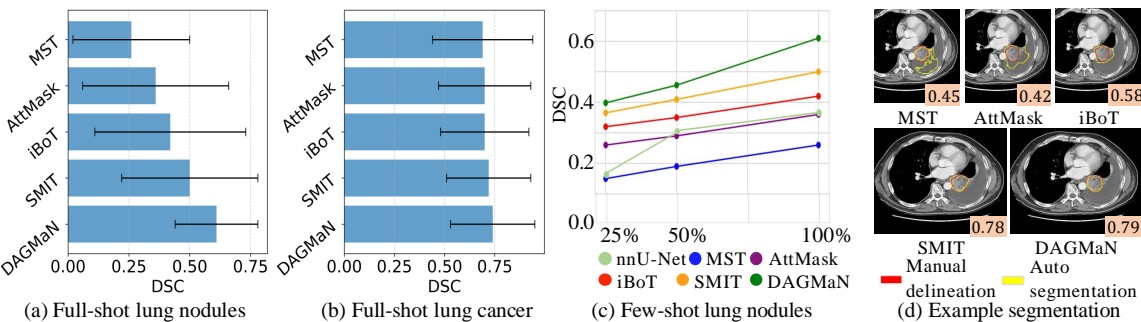

Figure 3: Full-shot segmentation for (a) lung nodules, (b) tumors, as well as (c) few-shot performance for lung nodules. (d) shows a representative patient with DSC accuracies.

Figure. 6 depicts the impact of placing SA module in the different stages of the Swin encoder. As shown, SA placed in stage #3 resulted in the highest accuracy with both linear probing and fine-tuning for classifying malignancy of lung nodules. Similarly, SA placed in stage #3 also resulted in highest accuracy for segmenting the lung nodules (DSC of 0.9 with FT, DSC of 0.7 with LP) compared to all other stages (second highest AUC of 0.7 for stage #4 with FT). SA placed in stage #3 also resulted in the highest inter-cluster and lowest inter-cluster separation for Task 4 (Supplemental Figure 9).

Finally, we implemented DAGMaN in ViT and found that it achieved higher accuracy for Task 1 (AUC of 0.66 with LP and 0.847 with FT) compared to random masking (AUC of 0.643 with LP and 0.816 with FT) as shown in Supplemental Table 2.

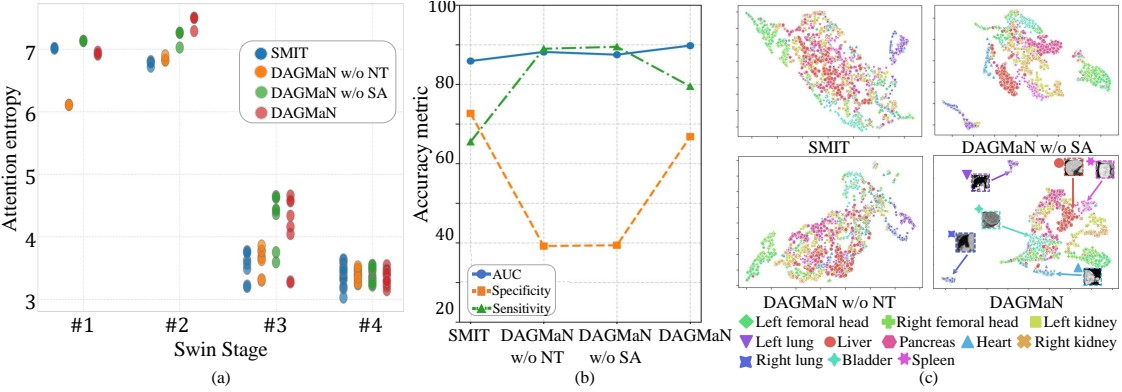

Figure 4: Impact of attentive masking and noisy teacher co-distillation. (a) shows attention-head entropy across the Swin stages with different masking strategies on the LIDC dataset. (b) shows the impact of the same strategies on lung nodule malignancy classification and (c) shows differences in unsupervised clustering of distinct organs using pretrained features.

### 4.7. DAGMaN learns better attention maps

We analyzed the capability of DAGMaN to extract attention maps for classification task 3 involving predicting immunotherapy outcome. DAGMaN was evaluated against AttMask (Kakogeorgiou et al., 2022) as well as without noisy teacher. Our analysis showed that DAGMaN

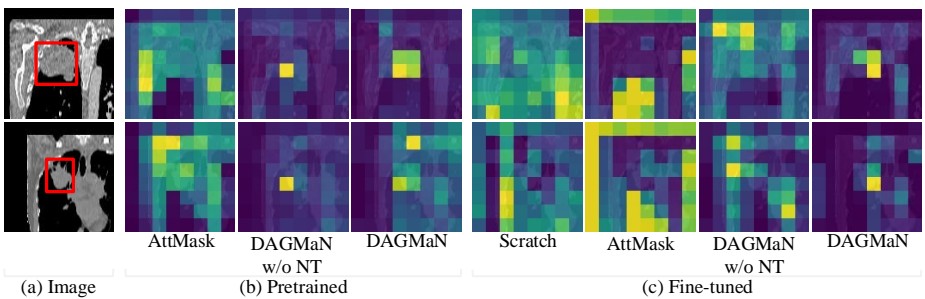

Figure 5: Attention maps for representative examples from the immunotherapy response prediction dataset. (a) shows CT image slice. Attention maps are shown for (b) semantic attention layer pretrained with AttMask, DAGMaN w/o noisy teacher and DAGMaN, (c) semantic attention layer fine tuned with pretrained weights, including training from scratch, AttMask, DAGMaN w/o noisy teacher and DAGMaN.

produced attention visualizations focused on the tumor both using pretrained features (Figure 5 (b)) and following fine-tuning (Figure 5 (c)). AttMask resulted in activations everywhere including lungs and chest, whereas noisy teacher co-distillation without semantic attention resulted in a dispersed attention, indicating importance of semantic attention.

## 5. Discussion and conclusion

Swin is a more accurate architecture than ViT for medical applications (Tang et al., 2022; Jiang et al., 2024; Cao et al., 2022). We introduced an approach to extract attention maps from Swin, which for the first time allows attention guided masking as well as capability to visualize explanation of model's prediction for classification tasks. We also introduced a noisy teacher co-distillation approach, which shows that combining semantic attention guided masking with noisy teacher enhances the attention head diversity in the first three stages of Swin while increasing downstream accuracy in both full- and few-shot training regimes for image-level and dense-pixel prediction tasks. Our results are consistent with observations for natural images that also showed increased attention diversity with masked image modeling improves accuracy (Xie et al., 2023). DAGMaN produced robustly accurate segmentations independent of tumor size variations. Although we developed our approach for Swin, we found that DAGMaN was also applicable to ViT. DAGMaN though more resilient than other methods, was adversely impacted by tumors occurring in the mediastinum, as well as tumors fused with collapsed lung. Both of these are difficult conditions for segmentation. Future work will also focus on combining DAGMaN-based approach for parameter efficient fine-tuning by understanding the dynamics of feature refuse, which is beyond the scope of this work.

## 6. Acknowledgements

This research was partially supported by the NCI R01CA258821 and the Memorial Sloan Kettering Cancer Center Support Grant/Core Grant NCI P30CA008748

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

## Appendix A. Implementation details

Pytorch (Paszke et al., 2019) and MONAI (Cardoso et al., 2022) libraries were used for implementation and training of the various models. For pretraining, DAGMaN and various comparable methods such as AttMask (Kakogeorgiou et al., 2022), MST (Li et al., 2021), iBot (Zhou et al., 2022) and SMIT (Jiang et al., 2022) were pretrained on datasets as mentioned in Section 4.1.

Two different transformer structures were evaluated in this study, namely ViT and Swin. The ViT (Dosovitskiy et al., 2021) architecture comprised of 12 transformer blocks, 768 embedding features, and 8 multi-head self attention. The Swin (Liu et al., 2021b) architecture used a depth of [2,2,8,2] and [4,4,8,16] multi-head for each transformer depth, and a feature embedding size of 384. This setup also included a window size of $4 \times 4 \times 4$ and patch size of $2 \times 2 \times 2$.

We generated augmented views by re-sampling the scans at 2mm $\times$ 2mm $\times$ 2mm voxel spacing and then randomly cropping $128 \times 128 \times 128$ voxel scans. The networks were optimized using ADAMw (Loshchilov and Hutter, 2017) with a cosine learning rate scheduler (Loshchilov and Hutter, 2016) trained for 800 epochs with an initial learning rate of $8e^{-4}$ and warmup for 80 epochs. A path drop rate of 0.1 was applied to the student model, and pretraining was conducted on four NVIDIA A100 GPUs (each with 80GB memory). Hyperparameters $\lambda_{AITD} = 0.1$, $\lambda_{GITD} = 0.1$, amd $\lambda_{AMPD} = 0.1$ in Section 3.2 were determined experimentally via grid-search. Degenerate solutions were avoided using centering and sharpening operations (Caron et al., 2021; Jiang et al., 2022).

## Appendix B. Downstream task descriptions

### Task 1: Binary classification of Malignant and Benign lung nodules from patients screened for lung cancer (LC)

The Lung Image Database Consortium (LIDC) dataset (Armato III et al., 2011) consists of 1,010 patients from seven different institutions, with a total of 2,426 lung nodules extracted using pylidc library. Each nodule was rated on a malignancy from scale 1-5, with ratings of $1-3$ as benign and $4-5$ as malignant, following the approach of (Chen et al., 2019). This grouping approach resulted in 2,054 benign and 540 malignant nodules. We employed a 3-fold stratified cross-validation on 1624 nodules, and used the rest 1,000 as test set. We used a batch size of 40 with a learning rate of $2e^{-4}$ for 1,000 epochs on NVIDIA

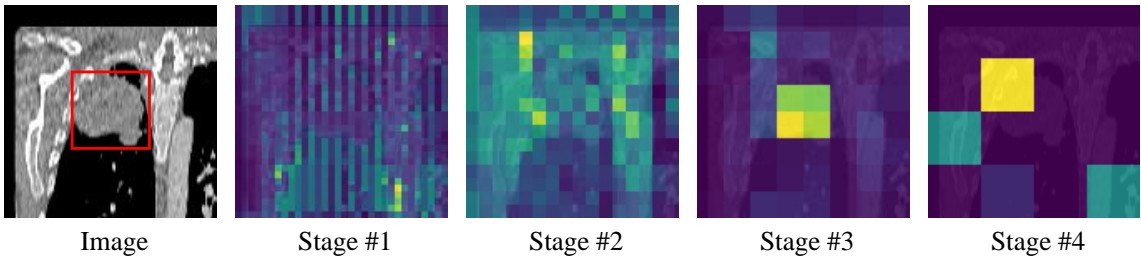

| Image | Stage #1 | Stage #2 | Stage #3 | Stage #4 |

Figure 6: The pretrained attention map when putting the semantic attention layers at different stage of Swin transformer.

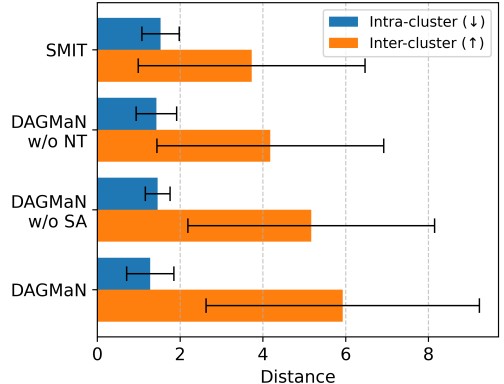

Figure 7: The impact noisy teacher and semantic attention on the inter- and intra cluster of the OrganMNIST dataset.

Table 1: Impact of losses on nodule malignancy classification

| AMIP | AMPD | AITD | GITD | AUC |
|------|------|------|------|-------|
| ✓ | ✗ | ✗ | ✗ | 0.859 |
| ✓ | ✓ | ✗ | ✗ | 0.869 |
| ✓ | ✓ | ✓ | ✗ | 0.878 |
| ✓ | ✗ | ✓ | ✓ | 0.865 |
| ✗ | ✓ | ✓ | ✓ | 0.829 |
| ✓ | ✓ | ✓ | ✓ | 0.895 |

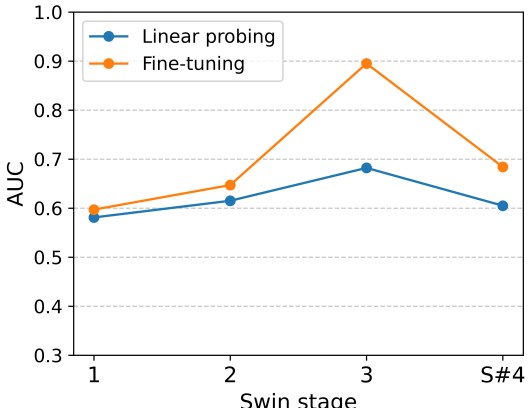

Figure 8: Impact of semantic attention module placement on nodule malignancy classification

Figure 9: Impact of semantic attention module placement on unsupervised organ clustering

A40 for fine-tuning and evaluated the models using the Area Under the Curve (AUC) metric.

**Task 2: Lung tumor and nodule segmentation**

For segmentation, we used 377 3D CT scans from lung cancer (LC) patients before radiation treatment, sourced from the publicly available cancer imaging archive (TCIA-LC) (Aerts et al., 2015) dataset. We fine-tuned our model on this dataset (350 for training and 27 used for validation), and then tested on the LIDC containing lung nodules (Armato III et al., 2011). As this data represents nodules that are typically small compared to malignant cancers, results for nodules larger than 3 cc were reported. In addition, the model was also evaluated in an institutional dataset of 200 patients with advanced non-small cell lung cancers (same patients used in Task 3 below).

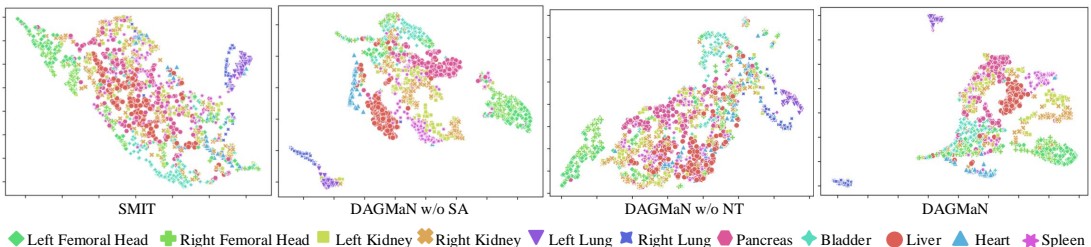

Figure 10: Differences in unsupervised clustering of distinct organs using pretrained features from different configurations.

Table 2: DAGMaN generalized to both ViT and Swin transformer.

| SSL Task | Network | Nodule malignancy classification | | Unsupervised organ clustering | |
| --- | --- | --- | --- | --- | --- |
| | | Linear probing | Fine-tuning | Intra-cluster ($\downarrow$) | Inter-cluster ($\uparrow$) |
| SMIT | ViT | 0.643 | 0.816 | $1.30 \pm 0.51$ | $6.44 \pm 5.27$ |
| DAGMaN | ViT | 0.662 | 0.847 | $1.15 \pm 0.41$ | $6.59 \pm 4.60$ |
| SMIT | Swin | 0.629 | 0.859 | $1.53 \pm 0.45$ | $3.73 \pm 2.74$ |
| DAGMaN | Swin | 0.642 | 0.895 | $1.28 \pm 0.57$ | $5.93 \pm 3.30$ |

## Task 3: Binary prediction of immunotherapy response

We defined response no disease progression $\geq 6$ months after start of immunotherapy treatment start vs disease progression within 6 months as non durable benefit [NDB]). Pretreatment CT scans of 200 patients with non-small cell lung cancers were used. Patients were scanned either with contrast or non-contrast scans and acquired using lung reconstruction kernel. The response distribution of Durable Clinical Benefit [DCB] to Non-Durable Benefit [NDB] is 82:118. Three-fold stratified cross-validation was applied. We used a batch size of 40 with a learning rate of $2e^{-4}$ for 1,000 epochs on NVIDIA A40 for fine-tuning and evaluated the models using the Area Under the Curve (AUC) metric.

## Task 4: Unsupervised clustering of organs from OrganMNIST

We evaluated our pretrained model's capability to differentiate various organs on the OrganMNIST3D dataset, which consists of 1,743 3D CT scans representing 11 different organs in the abdomen. The performance was evaluated using inter-cluster and intra-cluster distances, computed using the Euclidean distance.

