# OpenReview forum: "Co-distilled attention guided masked image modeling with noisy teacher for self-supervised learning on medical images"
_MIDL.io/2025/Conference — MIDL 2025 Poster_

### Official Review · Reviewer_We8X · 2025-02-22

**Confidence:** 4
**Preliminary Rating:** 4
**Recommendation:** Poster
**Final Rating:** 3

**Summary:**

Authors present an attention guided masking strategy for
self-supervised pre-training of the swin architecture. They define a
class token ([CLS]) in the swin architecture, use this token for
computing attention across patches, which is later used for masking,
and train the architecture using a student-teacher model. Experiments
show favorable results compared to other self-supervised training
strategies on multiple downstream tasks.

**Strengths:**

+ Combination of attention guided masking strategy and student-teacher
  approach is interesting and novel to the best of my knowledge. Using
  a teacher network instead of an auxiliary network to guide the
  attention for patch mask selection is an intriguing direction.
+ The swin architecture is an important architecture for medical image
  computing. I appreciate the effort by the authors to design a
  solution for this architecture.
+ The experimental design is well designed with appropriate
  comparisons.

**Weaknesses:**

+ Enforcing consistency between teacher and student representations is
  a crucial component of this approach. However, the computation of
  multiple losses and at different stages seem to be arbitrarily
  chosen. For instance, the reasoning behind computing one after the
  3rd stage and the other after the 4th stage is not clear.
+ Furthermore, it is not clear to me why one needs both GITD and
  AMPD. They seem to serve similar purposes. Or perhaps this needs to
  be clearly explained. On the same point, explanation of these losses
  needs to be improved. How is the average pooling applied? How are
  the patch distributions computed - does this loss sum over all the
  patch tokens?

**Detailed Comments:**

+ Figure 1D is quite important, yet I think it is not clear at
  all. The method explanation also relies on this figure. I suggest
  improving this figure to make the explanation more clear. For
  instance, the SA module can be better explained in this figure.
+ The choice of consistency losses can be better explained.

**Justification Of The Final Rating:**

Authors' explanations raised more questions in my mind regarding the choices. I feel like the algorithmic choices are based on accuracy improvement, especially considering the 2nd answer, rather than algorithmic intuition.

**Justification Of The Preliminary Rating:**

The article has novelty and experimental results are backing up the
concept. However, choices in implementing the main idea is not well
justified in my opinion. More specifically, authors need to provide
better justifications for all the consistency losses they utilize.

**Questions To Address In The Rebuttal:**

+ Please better justify the choices of multiple consistency losses. In
  this justification, I believe it is important to provide algorithmic
  / intuitive / mathematical justifications rather than retrospective
  analysis based on the experimental results.

**Special Issue:**

No

---

> ### Author Response · Authors · 2025-03-07
> **Rebuttal**
>
> Dear Reviewer We8X,
>
> Thanks for all your comments. We have now explain the consistency of the loss, the differences of GITD and AMPD, including more details about Semantic module in Figure 1D.
>
> ***Our answers to the weaknesses and other stated concerns are below:***
>
> **1. consistency of the loss**:  There are 4 different losses in the proposed method, including AMPD, AMIP, AITD and GITD. Among them, AITD and AMPD are implemented in the stage #3 while AMIP and GITD are implemented in stage #4.  AITD or attention guided image token distillation is computed to measure the dissimilarity of the [CLS] token injected into the semantic attention and it can only be computed after stage #3. AMPD or attention guided masked patch distillation is computed to measure the dissimilarity of patch token embeddings computed by teacher and student network, particularly emphasizing the differences in the extracted embeddings within the attention guided masked regions. Attention masks are computed as a result of semantic attention. Hence, this loss is computed following stage #3.
> On the other hand, the AMIP or attention guided masked image prediction is computed to assess the difference in the generated image patches in the masked regions and hence is most meaningful to be computed at the end of stage #4. Similarly, the GITD or global image token distillation computes the dissimilarity of the token embeddings computed for the whole image views, and is thus computed after stage #4, which best extracts the global image semantics.
>
> **2. Why need GIDT and AMPD**: Both GITD and MPD are important as shown in the ablation study result in Supplementary Table 1, where including both of these losses in addition to others produced best accuracy. On the other hand, removing either of these resulted in a lowering of accuracy. GITD measures the dissimilarity of global embeddings extracted from corrupted image views ($\hat{v}$ and $\tilde{u}$). GITD was computed with the final set of token embeddings from the last Swin transformer block followed by an average pooling layer. This process computes the mean of all token embeddings in the spatial dimension, producing a global feature representation of the image.
> On the other hand, AMPD only focuses on the differences in the extracted features within the attention-guided masked regions to ensure similar features are computed by the student and teacher networks. Hence, the patch embedding differences are averaged over masked patches only. Both of these losses are essential to extract different aspects of summarizing images using the token embeddings.
>
> **3. Figure 1 D improvement**: We have improved the Figure 1D with detailed semantic attention module.

---

### Official Review · Reviewer_3fQJ · 2025-02-24

**Confidence:** 4
**Preliminary Rating:** 4
**Recommendation:** Poster

**Summary:**

This paper addresses the challenge of improving self-supervised learning (SSL) for medical image analysis through masked image modeling (MIM). Aiming at the problem that traditional MIM methods lead to information leakage and SSL simplification, this paper proposes a framework that performs co-Distillation Attention Guided MAsking with Noisy teacher (DAGMaN). By architecturally enhancing Swin for performing attention-guided MIM and the noisy teacher, they demonstrate that DAGMaN produces global self-attention and achieves higher attention head diversity, which is crucial for downstream task performance.

**Strengths:**

1. The paper proposes an attention-guided masking mechanism with noisy teacher (DAGMaN) within a co-distillation learning framework that selectively masks semantically important regions, reducing information leakage and increasing SSL pre-training difficulty. This is enabled by modifying the Swin transformer to extract global attention, making attention-guided masking feasible and expanding its applicability in medical imaging.
2. This paper introduces a noisy teacher in the co-distillation framework to enhance the diversity of attention heads and thus improve the performance of downstream tasks, since feature diversity is crucial for accurate diagnosis and segmentation.
3. The paper provides extensive experiments across multiple medical imaging tasks, demonstrating the versatility and robustness of DAGMaN. The results show consistent improvements in both full-shot and few-shot learning scenarios, highlighting the method's practical applicability.
4. The paper is well-structured and clearly written, making it easy to follow and understand.

**Weaknesses:**

1. Figure 1 lacks sufficient detail in visually representing the DAGMaN framework, making it hard to interpret. Adding descriptive annotations or a step-by-step textual explanation would greatly enhance readability and clarify the framework's structure and flow.
2. While the paper modifies Swin by adding a semantic attention (SA) module at Stage #3, it does not thoroughly explain why Stage #3 was selected over other stages (e.g., via ablation studies on SA placement).
3. Despite emphasizing its role in preserving attention diversity, the paper does not provide dedicated ablation studies to isolate the impact of this component.
4. Although the paper highlights improved performance, it does not rigorously analyze cases where DAGMaN underperforms or fails. For example, no detailed discussion is provided for segmentation errors in irregular tumors or classification failures in rare pathologies, which are critical for clinical applicability.

**Detailed Comments:**

The relationship between the student and teacher networks in Figure 1 is not well-illustrated. Additionally, while the text mentions the use of co-distillation losses (AMPD, AITD, and GITD), Figure 1(D) does not visually represent GITD, creating a slight disconnect between the description and the diagram.

**Justification Of The Preliminary Rating:**

This paper presents DAGMaN, an attention-guided masking mechanism with a noisy teacher in a co-distillation framework, enhancing SSL pre-training by reducing information leakage and increasing attention diversity. Extensive experiments across multiple medical imaging tasks demonstrate consistent performance gains, but the paper lacks detailed justifications for SA module placement, ablation studies on attention diversity, and error analysis on challenging cases. Despite these limitations, the novel approach and strong results make this a valuable contribution, warranting a Weak Accept.

**Questions To Address In The Rebuttal:**

refer to Weaknesses and Comments

---

> ### Author Response · Authors · 2025-03-07
> **Rebuttal**
>
> Dear Reviewer 3fQJ,
>
> Thank you for the comments. We have provide sufficient details in the DAGMaN framework, explaining selection of stage 3, including ablation studies on noisy teacher, discussing where DAGMaN underperforms or fails.
>
> ***Our answers to the weaknesses and other stated concerns are below:***
>
> **"1. Figure 1 lacks sufficient detail in visually representing the DAGMaN framework ..."**:  We have clarified the details and added details and included explanation of the acronyms to help improve the readability.
>
> **2. why Stage #3 was selected over other stages**: We conducted ablation experiments on the placement of semantic attention or SA module to study both localization of the attention (Supplementary Figure 6) as well as accuracy of lung nodule classification (Supplementary Figure 8) and unsupervised clustering based differentiation of organs (Supplementary Figure 9). Our analyses clearly showed superior performance when SA was placed after Stage 3.  The reason for improved performance when placing SA after stage 3 is as follows. Stages 1 and 2 extract low level features and lack sufficient global anatomical context in the extracted features due to the hierarchical representation used in Swin, which requires more layers to extract global context. Features extracted after Stage 4 have sufficient global context but have very low resolution, which is insufficient to extract fine-grained spatial attention. Hence, we used Stage 3 to place the semantic attention as it balances the anatomic context extracted by the features and the image resolution. Also, the number of token embeddings produced after stage 1 and 2 would be prohibitively large for computation with reasonable computational resources.
>
> **3. Ablation of noisy teacher on preserving attention diversity**: We included a figure showing how the noisy teacher increased attention head diversity on the LIDC dataset on figure (4) a.  We also showed the impact of noisy teacher on LIDC prediction accuracy on figure (4) b; we also showed how the noisy teacher impact both pre-training and fine-tuning attention map in figure 5; Finally we showed how the noisy teacher impact the clustering results in ORganMinst3D.
> A more thorough and detailed analysis would be beyond the scope due to page restrictions and would make it difficult for us to capture everything cohesively. Reviewer’s point is well taken and we are including more detailed analysis of this aspect in an extended journal publication.
>
> **4. Discuss where DAGMaN underperforms or fails**: We have included a description of where the model underperforms and fails as limitations. Specifically, we found that DAGMaN segmentation underperforms in cases of hyperintense lung nodules or tumors that merge with the mediastinum. For classification, the model struggles with giant tumors of irregular shape that are fused with or in close proximity to the mediastinum, particularly in cases of lung collapse. Future work will focus on incorporating multi-modality data, such as PET imaging and clinical text information, to improve both segmentation and classification accuracy.
>
> **5. Details in Figure 1**. We have clarified figure 1 regarding on the detailed method and also included GITD in Figure 1 (d)

---

### Official Review · Reviewer_ftvt · 2025-02-28

**Confidence:** 4
**Preliminary Rating:** 4
**Recommendation:** Poster
**Final Rating:** 4

**Summary:**

The authors introduce a self-supervised training pipeline based on Swin transformers. Specifically, they use attention-guided masking, inside a (noisy) teacher-student framework. They propose modifications to the Swin architecture, to be able to mask based on global attention. In a large number of experiments, they show that their approach works better than existing ViT-based approaches.

The paper and experiments make a convincing case for the efficacy of the authors' approach, but since there are many elements introduced simultaneously, it is a bit hard to take away clear insights from the paper.

**Strengths:**

- The authors perform a lot of work for an 8-page paper, including comparing multiple architectures, ablation experiments, and experiments on a wide variety of downstream tasks: classification, segmentation and clustering.
- Performance metrics are suitable for all tasks, and datasets are handled properly in terms of cross-validation/test sets.
- The proposed architecture changes are minimal, but seem effective from the results, so this could be a nice insight.

**Weaknesses:**

- The amount of work done for this paper makes it a slight weakness in terms of readability: the main message of the paper is hard to parse. The authors tried to present a lot (and it seems also played around with whitespace etc. to make this possible :)).
- Connected to this: section 3 introduces many methodological aspects, including architecture changes and loss terms. It is unclear which of those are novel, and which the authors use directly from literature.
- While there are quite some ViT baselines included, I think it would be nice to have some simpler baselines as well, such that the strength of the proposed method can be placed in a bit wider that Swin vs. ViT context. For instance:

- Include task-specific baselines. E.g. for segmentation, you could have included nnU-Net trained on 25, 50, 100% of data for fig (c). That would practically be my first choice if I want to do that task, so it is nice to see how the authors' method compares. Similarly for classification you could train a simple convolutional classifier. Of course, these are not as flexible as an SSL method, but they do frame the performance numbers for the tasks better.

**Detailed Comments:**

- Some figures have really small font, and would be unreadable in a printed paper. Especially Fig 3/Fig 4. I think this is related to showing a lot of results, but where possible it would be nice to make those more readable.
- p3, on the top it still says 'Short Title'

**Justification Of The Final Rating:**

The authors introduce a novel training setup for self-supervised learning, based on Swin transformers and co-distillation. They introduce several new components to make this work and show a breadth of experiments to demonstrate that their setup improves performance across a range of tasks. I think this will be an interesting contribution for the conference.

**Justification Of The Preliminary Rating:**

The authors introduce original architecture modifications to Swin to allow for attention-guided self-supervised learning, which is a nice contribution. While the paper is a bit hard to read, mainly because the authors try to present a lot of work in little space, I think this would be a nice contribution to the conference. Especially if the main points can be presented clearly!

**Questions To Address In The Rebuttal:**

- Can you indicate which parts of the methodology are original (and unique to DAGMaN)?
- Can you include either task-specific baselines, or typical task-specific performance measures for the various tasks?

**Special Issue:**

Yes

---

> ### Author Response · Authors · 2025-03-07
> **Rebuttal**
>
> Dear Reviewer ftvt,
>
> Thank you for the comments. We have clarified the novelty and the explanation of methods in response to the reviewer’s suggestions.
>
> ***Our answers to the weaknesses and other stated concerns are below:***
>
> **1. Readability of the paper**:  We performed a thorough and rigorous analysis to understand the applicability and limitations of our approach to segmentation and classification tasks – as the reviewer pointed out, this required several experiments. We went through the paper and improved the clarity of the figures and tried to add more explanations to improve the clarity and readability of the paper to the best of our ability.
>
> **"2. It is unclear which of those are novel, and which the authors use directly from literature"** We further clarified our innovation and contributions, which include the introduction of the semantic attention module that enables Swin to conduct global self-attention and extract the global attention map that is required to perform attention guided masking. We also introduced a noisy teacher where the teacher network processes noisy images created by masking through random patch dropout for self-supervised learning performed between an exponentially moving averaged teacher and student network. Prior works[1] used noise introduced through dropout of all feature layers within a knowledge distillation framework that uses a larger teacher network to refine a small student network applied to classification tasks involving natural images. Another prior work[2] related to medical image segmentation applied mask dropout to images combined with signed distance maps processed by a linear projection head of a convolutional network. We, on the other hand, introduced a noisy teacher that processes masked input tokens to regularize the self-distillation losses involving global and local patch tokens extracted by the two networks.
>
> **3. CNN baselines**  We have included a nnU-Net baseline for LIDC segmentation also a 3D ResNet for immunotherapy and nodule malignancy prediction.
>
> **4. Figure font size**: We have increased figure font size for better readability, including figure 1,2,3,4
>
> **5. p3 ... 'Short Title'**: We have fixed this in the updated submission.
>
> ***Our response to the concerns in the rebuttal are:***
>
> **1. Method novelty (and unique to DAGMaN) ".** Please see our response for the weakness.
>
> **2. task-specific baselines.**  We have included a nnU-Net baseline for LIDC segmentation also a 3D ResNet for immunotherapy and nodule malignancy prediction
>
> **References**
>
> [1] Lee, Hyoje, Yeachan Park, Hyun Seo, and Myungjoo Kang. "Self-knowledge distillation via dropout." Computer Vision and Image Understanding 233 (2023): 103720
>
> [2] You, Chenyu, Yuan Zhou, Ruihan Zhao, Lawrence Staib, and James S. Duncan. "Simcvd: Simple contrastive voxel-wise representation distillation for semi-supervised medical image segmentation." IEEE Transactions on Medical Imaging 41, no. 9 (2022): 2228-2237

---

> > ### Comment · Reviewer_ftvt · 2025-03-11
> >
> > Thanks for all the updates! And also great that you were able to include the supervised baselines.
> >
> > My final question is on nnU-Net performance. The reason I was curious to see its performance in Fig 3, is that the 100% supervised performance seemed rather low for segmentation models. Now nnU-Net achieves less than 0.4 DSC when trained on the full training set?
> >
> > I cannot find an exact application of nnU-Net to LIDC-IDRI, but for instance this ref [1] shows multiple CNN baselines achieving DSC around 0.8.
> >
> > Could you explain why the fully supervised model's performance is so low?
> >
> > [1] Y. -A. Chen et al., "Lung Nodule Segmentation in LDCT: Modified 3D nnUNet with Unified Focal Loss," 2023 International Conference on Electrical, Computer and Energy Technologies (ICECET), Cape Town, South Africa, 2023, pp. 1-8, doi: 10.1109/ICECET58911.2023.10389288.

---

> > ### Author Response · Authors · 2025-03-11
> > **Response to Reviewer ftvt comment**
> >
> > Thank you for your question and the suggestion about the paper using nnU-Net on the LIDC dataset. There are several key differences between our and their paper.
> > Foremost, our goal was to assess the out of distribution robustness performance of the deep learning methods trained using malignant lung cancers from the public dataset containing non-small cell lung cancers (NSCLC) and evaluate on the open source LIDC dataset that contains benign, pre-cancerous and early cancerous lesions. Hence, all methods including DAGMaN and nnU-Net were trained on identical NSCLC cancers and evaluated on the LIDC dataset. On the other hand, the referenced paper trained nnU-Net on the LIDC dataset and evaluated the model on the same LIDC dataset, testing on in-distribution dataset.
> >
> > Second, we used the standard nnU-Net with an input size of 128 x 128 x 128 pixels. This image size was selected to cover most of the lung and use the same image size across all evaluated methods.
> >
> > Third, the referenced paper employs a modified nnU-Net architecture incorporating Dilated-Res2Block and Gated Attention to enhance feature extraction and skip connections within the traditional U-Net structure. It also utilizes a 64x64x64 patch input size and an Asym-Unified Focal Loss to address the highly imbalanced data distribution in the lung nodule dataset. This approach essentially alters the network from nnU-Net to a custom approach developed by the authors. We, on the other hand, use the standard nnU-Net.
> >
> > All these reasons contributed to a lower accuracy of nnU-Net compared to what was reported in that paper. This by no means makes nnU-Net that we used any worse. The key differences were that we evaluated capability of all methods to handle the more challenging out-of-distribution data (also relevant in clinical applications), a larger field-of-view size (more realistic in clinical applications), and faithfully adhered to the open-source nnU-Net architecture.

---

> > > ### Comment · Reviewer_ftvt · 2025-03-12
> > >
> > > Thanks for clarifying that, I missed that you trained on a different dataset to test out-of-distribution performance. Then the performance makes more sense for sure.

---

### Comment · Reviewer_3fQJ · 2025-02-22
**Request for Extension on Review Submission**

Dear AC,

I hope this email finds you well. I would like to request an additional three days to complete my review. I appreciate your understanding and consideration.

Best,
Reviewer 3fQJ

---

> ### Comment · Area_Chair_ADXf · 2025-02-22
>
> Dear Reviewer, thank you for the update - OK for the extension.
>
> If it looks like you won't be able please let me know so I can look for emergency reviewers.
>
> Best regards

---

### Comment · Area_Chair_ADXf · 2025-02-22
**0 reviews, expecting 1 in 3 days**

Dear PC,

As you see I have 0 reviews for this one so far. One reviewer already said he could not complete, one has requested an extension of 3 days so I am counting on this one.

I sent a second reminder to the other two from which I have not heard- one (senior) I believe will complete the review, the other one (PhD student) I don't know.

If no reply from the two, I'll wait until Monday morning before inviting 2 emergency reviewers - sounds OK ?

When will you release the list of emergency reviewers?

Best regards

---

### Comment · Area_Chair_ADXf · 2025-03-04
**rebuttal process**

Dear Authors,

I encourage you to actively participate in the rebuttal process, with a hard deadline on March 7 23:59 AoE.

You can leave official comments on OpenReview to address each reviewer's points separately, and upload a revised manuscript using the “rebuttal” function in OpenReview. The revision can include any additional details, experiments, or images that might be required in the paper within the page limit (max 9 excluding references, acknowledgements, and appendices), as well as other supporting documents in the rebuttal stage. Any changes must be highlighted in the revised manuscript.

Thank you

---

> ### Author Response · Authors · 2025-03-07
> **Response to AC**
>
> Dear Area Chair ADXf04,
>
> We thank the opportunity to present our rebuttal and the reviewers for their insightful comments and suggestions. In response to the suggestions, we have made the following changes.
>
> **(1)** further clarified our contributions.
>
> **(2)** including a nnU-Net base line for lung nodule segmentation and 3D ResNet for lung nodule malignancy prediction and immunotherapy outcome prediction.
>
> **(3)** increasing figure fonts for figure 1,2,3,4
>
> **(4)** providing sufficient details in DAGMaN framework in figure 1, providing details of Semantic Attention model in figure 1D
>
> **(5)** explaining choosing placement of Semantic Attention after Swin Sage 3
>
> **(6)** present ablation of noisy teacher on preserving attention diversity
>
> **(7)** discussing where DAGMaN fails
>
> **(8)** explaining loss consistency of different loss
>
> **(9)** explaining why both GITD and AMPD are needed and providing details on how the average pooling in GITD and how patch distributions computed in AMPD
>
>
> All the revisions are highlighted with blue text in the revised manuscript. We believe our manuscript is much improved from the prior version and hope the reviewers, AC, and PC would find the manuscript interesting for publication in MIDL conference.

---

### Author Rebuttal · Authors · 2025-03-08

**Rebuttal:**

We thank the reviewers for insightful comments. All three reviewers raised concerns regarding clarifying the novelty as well as improving the presentation of the figures especially related to the methodology. Accordingly, we improved Figure 1 describing the method, and clarified our novelty. We also improved the clarity of all other figures by increasing the font sizes as requested by reviewers.

Our methodological contribution includes a semantic attention module implemented into the Swin transformer, which enables the Swin to extract a global attention. The global attention is used by an exponentially averaged teacher to conduct attention guided masking for masked image modeling based self-supervised learning. Additionally, we introduce a noisy teacher approach wherein the teacher network is provided with noisy input tokens produced through random patch dropout for co-distillation learning, which increases attention diversity. Prior methods used noisy teacher regularization with knowledge distillation of intermediate features or outputs in supervised classification or semi-supervised learning frameworks.

As requested by Reviewer ftvt, we included additional CNN baselines using nnUnet and ResNet for segmentation and classification in few-shot settings.

Reviewer 3fQJ raised concern regarding the rationale for the placement of semantic attention and Reviewer We8X raised concern regarding the rationale for the various pretraining losses, particularly emphasizing the difference between AITD and GITD. We have added additional clarification in the methods and also included ablation experiments to assess the impact of placing semantic attention in the different network stages on classification accuracy, segmentation accuracy, as well as unsupervised clustering. These results are included in the supplemental figures due to space limitations in the main paper. Also, we included ablation experiments to assess the impact of the various losses, also included in the supplementary results.

Reviewer 3fQJ also asked for more thorough analysis of how the noisy teacher enhances attention diversity. We have clarified our experiments and included results of attention diversity computed using attention head entropy for all the stages in the paper for networks trained with and without noisy teacher, as well as with and without semantic attention.

Finally, we added additional clarification of instances where the DAGMaN approach fails to perform in the discussion.

**Supporting Material:**

/attachment/14eda5be409078c87d271cb8b5b5aac927509b2d.pdf

---

### Comment · Area_Chair_ADXf · 2025-03-10
**rebuttal discussion**

Dear Reviewers,

Please look at the authors' reply to your initial review. I encourage you to engage in discussion and consider the author replies, and update your rating and assessment (if justified) until the 14th of March.

Based on the initial review, the rebuttal discussion, and your final grades, I will write meta-reviews and propose a decision to the PC.

Thank you again for your very useful reviews.

---

### Meta-Review · Area_Chair_ADXf · 2025-03-19

**Recommendation:** Accept (Oral)
**Confidence:** 4

**Metareview:**

The paper proposes a modification of Swin transformers for self-supervision, using a student-teacher approach and a "semantic attention" module to compute a global attention map. The noisy teacher results in more diversity in attention heads.

In terms of strenghts, the reviewers appreciated the large amount of work performed including multiple architectures, ablation studies, and using several different tasks - classification, segmentation, and clustering; The architecture modifications, including the combination of guided masking and student-teacher approach were deemed novel and effective; and the relevance for medical imaging

In terms of weaknesses, the reviewers initially criticised the readability of the paper and figures; the motivation for the losses used; the lack of supervised, task-specific baselines to put performance in context; the lack of failure analysis

The authors thoroughly engaged in the rebuttal and modified the paper substantially, addressing reviewer concerns in detail. While some issues were not addressed fully, overall the authors put effort into the rebuttal.

In sum, the paper has a good balance of novel contribution, effectiveness, and evaluation, and helps refine a competitive architecture for the relatively low-data regime typical of medical imaging, thus increasing its chance of adoption.